

# Distribution and establishment of the alien Australian redclaw crayfish, *Cherax quadricarinatus*, in South Africa and Swaziland

Ana L. Nunes[1,2,3], Tsungai A. Zengeya[4], Andries C. Hoffman[5], G. John Measey[1] and Olaf L.F. Weyl[2]

[1] Centre for Invasion Biology, Department of Botany and Zoology, University of Stellenbosch, Stellenbosch, South Africa
[2] Centre for Invasion Biology, South African Institute for Aquatic Biodiversity, Grahamstown, South Africa
[3] Invasive Species Programme, South African National Biodiversity Institute, Kirstenbosch Research Centre, Cape Town, South Africa
[4] South African National Biodiversity Institute, Kirstenbosch Research Centre, Cape Town, South Africa
[5] Mpumalanga Tourism and Parks Agency, Nelspruit, South Africa

## ABSTRACT

**Background**. The Australian redclaw crayfish (*Cherax quadricarinatus*, von Martens), is native to Australasia, but has been widely translocated around the world due to aquaculture and aquarium trade. Mostly as a result of escape from aquaculture facilities, this species has established extralimital populations in Australia and alien populations in Europe, Asia, Central America and Africa. In South Africa, *C. quadricarinatus* was first sampled from the wild in 2002 in the Komati River, following its escape from an aquaculture facility in Swaziland, but data on the current status of its populations are not available.

**Methods**. To establish a better understanding of its distribution, rate of spread and population status, we surveyed a total of 46 sites in various river systems in South Africa and Swaziland. Surveys were performed between September 2015 and August 2016 and involved visual observations and the use of collapsible crayfish traps.

**Results**. *Cherax quadricarinatus* is now present in the Komati, Lomati, Mbuluzi, Mlawula and Usutu rivers, and it was also detected in several off-channel irrigation impoundments. Where present, it was generally abundant, with populations having multiple size cohorts and containing ovigerous females. In the Komati River, it has spread more than 112 km downstream of the initial introduction point and 33 km upstream of a tributary, resulting in a mean spread rate of 8 km year$^{-1}$ downstream and 4.7 km year$^{-1}$ upstream. In Swaziland, estimated downstream spread rate might reach 14.6 km year$^{-1}$. Individuals were generally larger and heavier closer to the introduction site, which might be linked to juvenile dispersal.

**Discussion**. These findings demonstrate that *C. quadricarinatus* is established in South Africa and Swaziland and that the species has spread, not only within the river where it was first introduced, but also between rivers. Considering the strong impacts that alien crayfish usually have on invaded ecosystems, assessments of its potential impacts on native freshwater biota and an evaluation of possible control measures are, therefore, urgent requirements.

Corresponding author
Ana L. Nunes, ananunes@sun.ac.za

## INTRODUCTION

Freshwater crayfish have been introduced globally, mostly for aquaculture and ornamental purposes, but generally their subsequent invasions have resulted in more ecosystem losses than benefits (*Lodge et al., 2012*). Continental Africa contains no native freshwater crayfish species, but three Australasian Parastacidae species, the Australian redclaw crayfish (*Cherax quadricarinatus*, von Martens), the smooth marron (*Cherax cainii* Austin and Ryan) and the yabby (*Cherax destructor* Clark), and a single North American Cambaridae species, the red swamp crayfish (*Procambarus clarkii* Girard), have been introduced (*Boyko, 2016*). All four species have been introduced into South Africa, but only *P. clarkii* and *C. quadricarinatus* seem to have successfully established wild populations (*Schoonbee, 1993*; *Van Rooyen, 2013*). Although *P. clarkii* has been introduced to several African countries and caused visible impacts (*Lowery & Mendes, 1977*; *Mikkola, 1996*; *Foster & Harper, 2006*), in South Africa the species is only known from a single locality and does not seem to be spreading (*Nunes et al., 2017*). Populations of *C. quadricarinatus* are more widespread in the country (*Du Preez & Smit, 2013*; *Van Rooyen, 2013*; *Coetzee et al., 2015*; *De Villiers, 2015*) and have also been reported from Swaziland (*De Moor, 2002*), Zimbabwe (*Marufu, Phiri & Nhiwatiwa, 2014*), Zambia and Mozambique (*Chivambo, Nerantzoulis & Mussagy, 2013*; *Nunes et al., 2016*). Globally, *C. quadricarinatus* has been translocated to non-native areas in Australia (*Doupé et al., 2004*; *Leland, Coughran & Furse, 2012*) and Indonesian territories (*Patoka et al., 2016*), and wild populations are also known from Israel (*Snovsky & Galil, 2011*), Jamaica (*Todd, 2005*; *Pienkowski et al., 2015*), Mexico (*Bortolini, Alvarez & Rodriguez-Almaraz, 2007*; *Vega-Villasante et al., 2015*; *Torres-Montoya et al., 2016*), Puerto Rico (*Williams et al., 2001*), Singapore (*Ahyong & Yeo, 2007*; *Belle et al., 2011*) and Slovenia (*Jaklič & Vrezec, 2011*). This species has also been introduced into several other countries (where wild populations do not exist) mostly due to its use in aquaculture (*Ahyong & Yeo, 2007*), but also due to being a very popular ornamental species that is readily available in the pet trade (*Belle et al., 2011*; *Chucholl, 2013*; *Patoka, Kalous & Kopecky 2014*).

*Cherax quadricarinatus* was first imported into South Africa in 1988 for research on its aquaculture potential, together with other *Cherax* species (*Van den Berg & Schoonbee, 1991*). Despite considerable interest in the aquaculture of this species in the late 1990s, its import and culture for commercial purposes has always been extremely restricted in South Africa. As a result, a farmer who failed to establish an aquaculture venture in South Africa around this time instead managed to successfully establish it in neighbouring Swaziland (*De Moor, 2004*). There are anecdotal reports that two batches of *C. quadricarinatus* were introduced from Australia to Swaziland, one for the abovementioned farm located near the Sand River Dam, close to the Komati River and the other to a farm near Manzini or Big Bend, in the Usutu River catchment (A Howland (general manager of *IYSIS* cattle ranch, inside which the Sand River Dam is situated), pers. comm., 2016). As a result of escape from captivity, crayfish spread to the Sand River Dam and later via the Sand River

into the Komati River (*De Moor, 2002*; *De Moor, 2004*; A Howland, pers. comm., 2016), where they were first detected in South Africa in 2002 (*De Villiers, 2015*). While there is no information on the outcome of the other aquaculture farm close to Manzini (in the Usutu River catchment), in 2012 *C. quadricarinatus* was detected in an outlet of Lake Nyamiti in the Ndumo Game Reserve (South Africa) (*Du Preez & Smit, 2013*), which eventually connects to the Usutu River, and in 2013 the species was being caught, sold and consumed in the villages bordering the Ndumo Game Reserve (*Coetzee et al., 2015*).

In June 2009, the species was also reported from a small wetland in a residential area close to Richard's Bay, in KwaZulu-Natal Province, South Africa (R Jones (Ezemvelo KZN Wildlife), pers. comm., 2016), a distant site, not directly connected to the initial introduction sites. This was probably the result of an escape or release via the aquarium trade, although data on the pet trade of this species in South Africa are not available.

Despite these initial reports of *C. quadricarinatus* in Swaziland and South Africa, no systematic survey has ever been carried out to determine their distribution, spread rate and population dynamics. This is of concern because crayfish invasions have generally been shown to result in strong impacts on recipient ecosystems (*Lodge et al., 2012*) and, given the absence of native crayfish on the African continent, these impacts are likely to be even stronger, especially upon native decapods, such as freshwater crabs from the genus *Potamonautes* (*De Moor, 2002*; *Jackson et al., 2016*; *Nunes et al., 2016*). In this study, we assess the current distribution, rate of spread and population dynamics of *C. quadricarinatus* populations in South Africa and Swaziland. In addition, for the Komati River (initial main river of introduction), we further investigate if population characteristics, such as abundance, biomass, sex ratio, body size and mass vary with distance to the introduction source, since traits of invasive populations have been shown to vary along invasion gradients (see review in *Iacarella, Dick & Ricciardi, 2015*).

## MATERIALS & METHODS

### Field study permissions

Permits for fieldwork in South Africa were obtained from the Mpumalanga Tourism and Parks Agency (MPB. 5523) and Ezemvelo KZN Wildlife (OP 4428/2015). For Swaziland, permission was granted from the Mbuluzi Game Reserve and All Out Africa Foundation.

### Study area

The study area was mainly situated in the Inkomati, Mbuluzi and Usutu River basins, all of which are international river systems running through Swaziland, South Africa and Mozambique. The Inkomati basin, mainly located in the Mpumalanga Province of South Africa, consists of three major sub-catchments, the Komati, the Crocodile and the Sabie-Sand (*Mpumalanga Tourism and Parks Agency, 2013*). The Komati sub-catchment is composed of the Komati River and its tributaries, one of which is the Lomati River. The Komati River rises in South Africa, west of Carolina in Mpumalanga, and flows for 480 km in a north-easterly direction through three countries (South Africa → Swaziland → South Africa → Mozambique). The Crocodile River is the main river in the Crocodile sub-catchment, originating north of Dullstroom and flowing eastwards towards its confluence
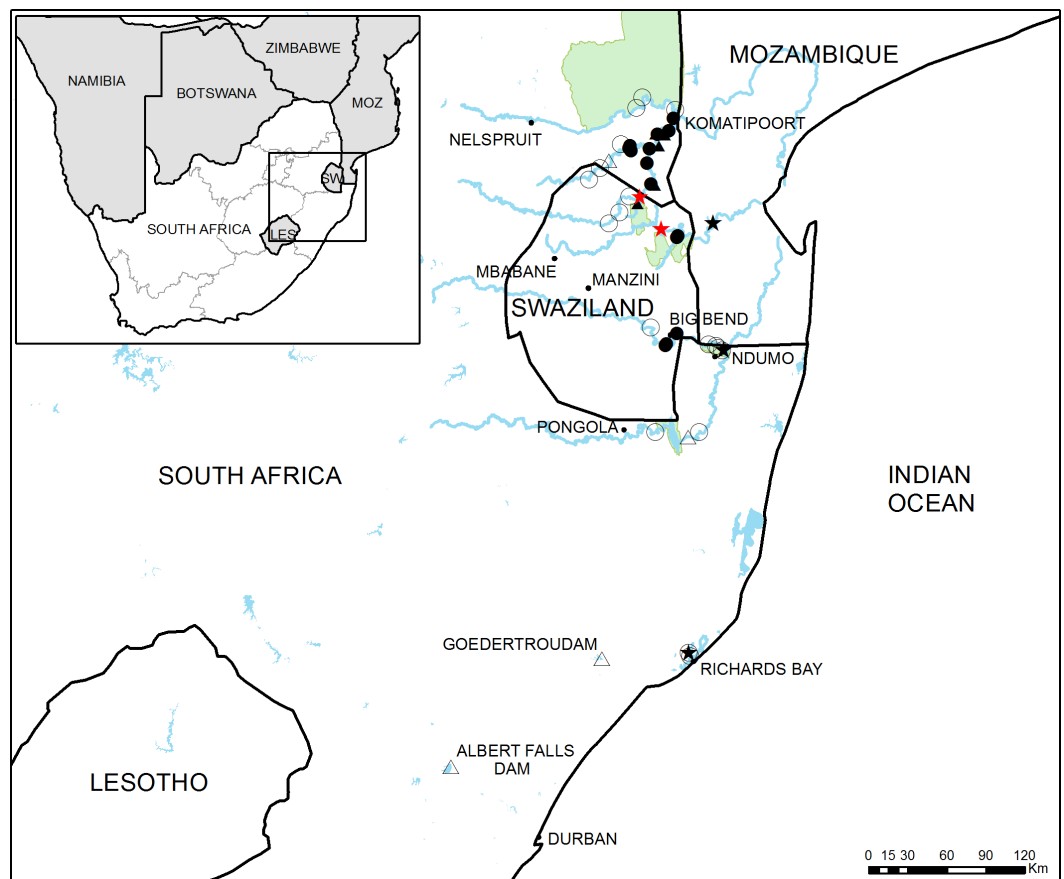

**Figure 1  Study area in South Africa and Swaziland.** General overview of the study area showing the 46 sampling sites used in this study. Full circles and triangles respectively represent river and dam sites where crayfish was found, empty circles and triangles represent river and dam sites where crayfish was not detected. Black stars indicate sites where crayfish presence has been previously reported and red stars represent the approximate potential points of first introduction.

with the Komati River. The Sand River Dam, where *C. quadricarinatus* was first introduced in Swaziland, is located in the Inkomati catchment (Figs. 1 and 2A).

The main river of the Mbuluzi basin is the Mbuluzi River, which originates in the Ngwenya hills in northwest Swaziland, close to the border with South Africa, and flows in an easterly direction through central Swaziland into Mozambique. At times, water is transferred from the Komati River basin to the Mbuluzi River basin via an intricate network of approximately 40 km of irrigation channels (A Howland, pers. comm., 2016; *Gustafsson & Johansson, 2006*). The Mlawula River, located close to the border with Mozambique, is one of its tributaries, which crosses several protected areas, such as the Mbuluzi Game Reserve and the Shewula Nature Reserve (Fig. 2B).

The Usutu River basin is bordered by the Mbuluzi and Inkomati River basins to the north and the Mhlathuze coastal catchment to the south. The Usutu, Pongola and Ngwavuma are its main sub-catchments. The main river of the Usutu sub-catchment is the Usutu River, which rises near Amsterdam, in Mpumalanga Province, and flows in a south-easterly

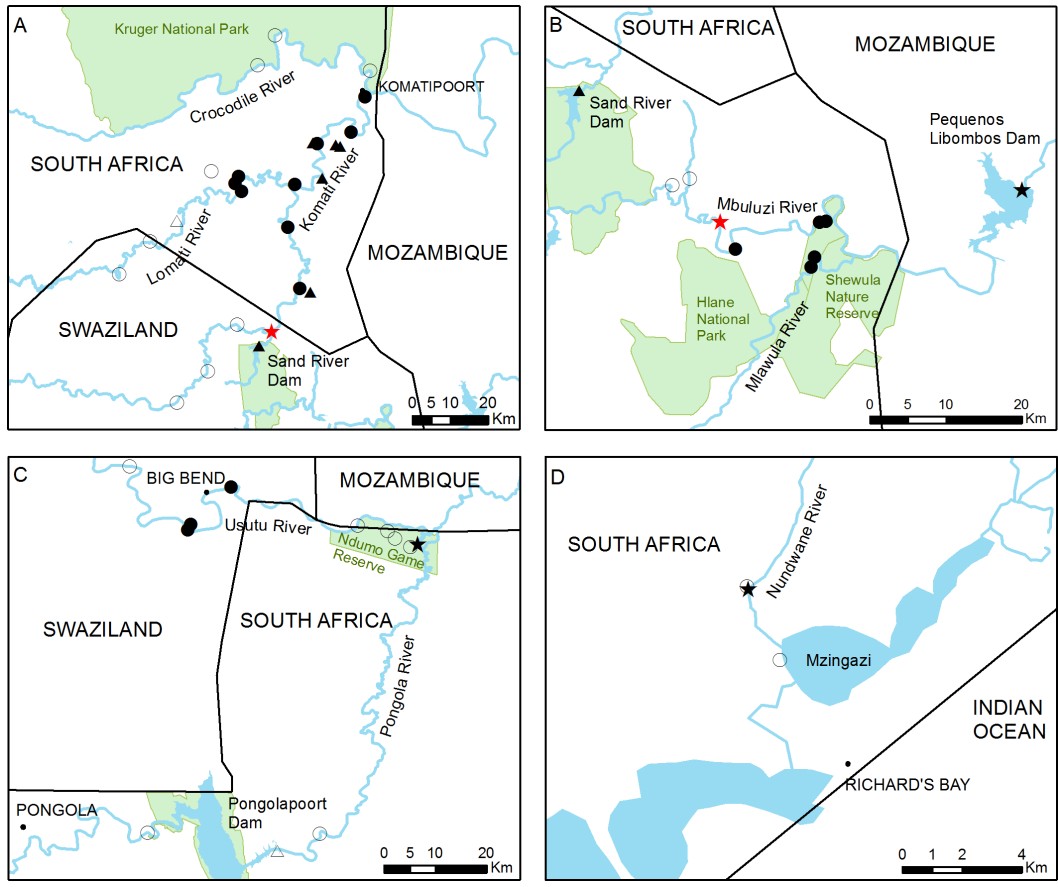

**Figure 2** **Detailed view of the four main study areas, with the 46 sampling sites surveyed in this study.** (A) The Inkomati, (B) Mbuluzi and (C) Usutu river basins and (D) Richard's Bay area. The approximate point of first introduction of *C. quadricarinatus* in the Komati River and the potential point of introduction in the Mbuluzi River are indicated with red stars. Full circles and triangles respectively represent river and dam sites where crayfish was found, empty circles and triangles represent river and dam sites where crayfish was not detected. Black stars indicate sites where crayfish presence has been previously reported.

direction through South Africa and Swaziland (*Beuster & Clarke, 2008*). It then emerges in the province of KwaZulu-Natal in South Africa where, for approximately 24 kilometres, it defines the border between this country and Mozambique, along the limits of the Ndumo Game Reserve. The Ndumo Game Reserve, a protected area characterised by numerous pans and wetlands, is crossed by the Pongola River, which rises in Northern KwaZulu-Natal, flows eastwards until the Pongolapoort Dam, from where it flows north-easterly to join the Usutu River in Mozambique (Fig. 2C).

Taking into account the reported sighting of *C. quadricarinatus* close to Richard's Bay, this area was also surveyed, as well as two large dams in the KwaZulu-Natal Province (Albert Falls and Goedertrouw Dams), where there have been unconfirmed records of crayfish presence (L Coetzer, pers. comm., 2015) (Figs. 1 and 2D).

## Sampling sites

A total of 46 sampling sites in different water bodies (main rivers, tributaries, pans, wetlands and dams) were surveyed between September 2015 and August 2016 (Fig. 1). Sampling sites were chosen by focusing on areas with suspected presence of *C. quadricarinatus*, according to published or grey literature and to personal communications from farmers, agriculture and conservation officials. Along the Komati River, which has a large number of weirs regulating its flow, nine sites were sampled, six downstream and three upstream of the initial introduction point (Fig. 2A). In contrast, the Lomati River is relatively less regulated and fewer sites (six) could be sampled on the main river or its tributaries due to difficult access. The three sampling sites on the Crocodile River were located upstream of its confluence with the Komati River and within the Kruger National Park (Fig. 2A). Sites on the Mbuluzi River and its tributaries were concentrated close to the Mozambican border, upstream (two) and downstream (four) of the potential point of introduction in this river (Fig. 2B). In the Usutu River, four points were sampled in Swaziland and one in South Africa. Three sampling points were selected in the Ndumo Game Reserve and two in the Pongola River, one upstream and one downstream of Pongolapoort Dam (Fig. 2C). In the Richard's Bay area, two points, one where crayfish were detected back in 2009 and one in a connected lake, were sampled (Fig. 2D). Finally, ten dams, most of which are primarily used to store water for agricultural irrigation, were also sampled.

Overall, 34 sites were sampled in lotic habitats, spaced at least 2.5 km from each other (but usually over 13 km), depending on where access to the rivers was possible. Survey sites in the rivers ranged between 100 and 150 m in length, depending on accessibility of the site. Twelve sites were sampled in lentic habitats. Each sampling site was surveyed at least twice (each site 2–4 times, except four sites where we could not return), once in the wet season (spring/-summer, September–March) and once in the dry season (autumn/winter, April–August), in order to confirm crayfish absences and detect differences in crayfish populations between seasons. The exception were three sites in the Crocodile River inside Kruger National Park, an area under strict jurisdiction of South African National Parks (SANParks) where, similarly to the four sites mentioned above, we could only sample once.

## Sampling procedure

At each of the sampling sites, visual observations of 5–10 minutes along the margins of the water body were made on arrival at the location, in order to look for crayfish specimens or moults. Subsequently, around ten (range: 3–15) ©Promar collapsible crayfish/crab traps (dimensions: 61 × 46 × 20 cm; mesh size: 10 mm), baited with approximately 100g of dry dog food, were set in the evening at each site, left overnight (14–16 h) and checked the following morning. The number of crayfish caught in each trap, as well as their cephalothorax length (to the nearest mm), mass (to the nearest g) and sex were registered. Crayfish abundance was calculated based on catch per unit effort (CPUE), per sampling session. Due to restrictions imposed by SANParks, traps could not be set in the Crocodile River, where instead electrofishing was conducted by wading for approximately 40 minutes per site, using a handheld SAMUS 725MP, with a 10 mm mesh scoop net.

## Data analysis

A chi-square goodness-of-fit test was used to test whether overall sex ratio, or per site and per season, was significantly different than the common sex ratio of 1:1 (e.g., *Bortolini, Alvarez & Rodriguez-Almaraz, 2007*; *Belle et al., 2011*). For the Komati River, we also investigated a possible relationship between each site's distance from the crayfish source of introduction (measured, in km, using Google Earth, downstream from the site of initial crayfish introduction and following the river's natural course) and crayfish catch per unit effort (abundance and biomass), sex ratio, size and mass. This was determined using Pearson's correlation coefficient or, when the assumptions of normality or homogeneity of variances were not met, the non-parametric Spearman rank correlation. The level of significance for all statistical tests performed was $p < 0.05$.

## RESULTS

### Presence/absence

*Cherax quadricarinatus* was detected in 22 out of the 46 sampling sites surveyed (Figs. 1 and 2, Table 1, Table S1). All sampling sites located on the Komati and Lomati rivers in South Africa had crayfish present, but no crayfish were detected in the upstream and more elevated sampling sites on both rivers in Swaziland. Crayfish were also found in the Mbuluzi River, but only in sampling sites downstream of the potential introduction point (interbasin transfer point between the Inkomati and Mbuluzi basins) in this river (Fig. 2B). Both sites on the Mlawula River, a tributary of the Mbuluzi River, also yielded crayfish. On the Usutu River, three sites close to Big Bend had crayfish, but crayfish were not caught further upstream in Swaziland, or downstream in Ndumo Game Reserve (Fig. 2C). Crayfish were also found in six out of the 12 sampled lentic habitats. However, they were not detected in the Crocodile and Pongola rivers, Ndumo Game Reserve and the Richard's Bay area (Fig. 2 and Table S1).

### Abundance

A total of 577 crayfish were caught during the wet season (383 males and 194 females), with a maximum of 63 individuals in a single trap (at site D01), whereas only 267 crayfish were caught in the dry season (149 males and 118 females). The maximum mass that a crayfish attained was 250 g, for an individual caught at site K06 (Table 1). In the Komati River, average crayfish abundances were quite high, ranging from 0.4 to 9.4 individuals trap night$^{-1}$ per site in the wet season and 1.0 to 7.0 individuals trap night$^{-1}$ in the dry season. High abundances were also found in dams (0.1–15.3 individuals trap night$^{-1}$), especially during the wet season. Abundances were lower in the Mbuluzi (1.0–4.5 individuals trap night$^{-1}$) and Mlawula rivers (0–4.0 individuals trap night$^{-1}$) and much lower in the Lomati and Usutu rivers ranging, respectively, from 0–0.7 individuals trap night$^{-1}$ and 0.1–0.8 individuals trap night$^{-1}$(Table 1). Average biomass was higher in the dry than in the wet season in the Komati (47.4 g trap night$^{-1}$ for dry season and 35.7 g trap night$^{-1}$ for wet season), Lomati (26.1 g trap night$^{-1}$ for dry season and 7.4 g trap night$^{-1}$ for wet season) and Mbuluzi rivers (27.8 g trap night$^{-1}$ for dry season and 22.0 g trap night$^{-1}$ for wet season). On the contrary, average biomass was higher in the wet than the dry season in the

Nunes et al. (2017), *PeerJ*, DOI 10.7717/peerj.3135

**Table 1  Attributes of the 22 sites where *C. quadricarinatus* was found.** Coordinates, location, elevation (m), distance to closest crayfish introduction point (km), season, catch per unit effort (CPUE, as number of individuals and biomass), average size (cephalothorax length, mm), average mass (g) and number of males and females, for each sampling site where crayfish was found. SD stands for standard deviation, M for males, F for females, SA for South Africa and SW for Swaziland.

| Site | Coordinates | Location | Elevation (m) | Distance to intro (km) | Season | CPUE (SD) (N/trap/night) | CPUE (SD) (g/trap/night) | Size M (SD) | Size F (SD) | Mass M (SD) | Mass F (SD) | M | F |
|---|---|---|---|---|---|---|---|---|---|---|---|---|---|
| K01 | 25°28″24.50″S 30°07″23.61″E | Komati River, SA | 130 | 112.11 | Wet | 2.2 (1.39) | 40.48 (26.42) | 57.17 (7.28) | 56 (9.59) | 50.83 (19.13) | 49.4 (22.49) | 12 | 10 |
| | | | | | Dry | 3.2 (5.07) | 33.01 (32.09) | 53.26 (19.35) | 50.97 (7.69) | 34.88 (26.29) | 32.93 (14.98) | 17 | 15 |
| K02 | 25°31″19.3″S 31°55″48.2″E | Komati River, SA | 153 | 97.26 | Wet | 3.1 (4.36) | 53.56 (53.39) | 66.07 (18.21) | 54.41 (9.47) | 77.57 (58.17) | 39.76 (21.34) | 14 | 17 |
| | | | | | Dry | 1.43 (1.81) | 39.89 (54.69) | 63.35 (19.82) | 46.63 (6.19) | 81.33 (62.52) | 32 (15.41) | 6 | 4 |
| K03 | 25°32″45.8″S 31°50″59.2″E | Komati River, SA | 174 | 81.96 | Wet | 9.4 (7.73) | 34.71 (16.73) | 50.08 (9.04) | 47.55 (7.79) | 30.65 (18.63) | 24.07 (12.55) | 65 | 29 |
| | | | | | Dry | 4.43 (4.96) | 66.4 (56.65) | 57.80 (14.20) | 49.21 (6.19) | 52.67 (43.58) | 28.63 (10.99) | 15 | 16 |
| K04 | 25°38″01.7″S 31°47″47.5″E | Komati River, SA | 198 | 61.11 | Wet | 0.38 (0.74) | 23.75 (44.01) | 72.33 (16.26) | – | 96 (64.09) | – | 3 | 0 |
| K05 | 25°43″29.4″S 31°46″49.8″E | Komati River, SA | 233 | 44.94 | Wet | 2.88 (5.49) | 32.42 (39.75) | 56.92 (14.69) | 46.55 (6.85) | 53.17 (42.59) | 25.09 (12.37) | 12 | 11 |
| | | | | | Dry | 7 (11.93) | 47.15 (32.96) | 63.35 (11.86) | 64.22 (5.11) | 62.27 (31.55) | 58.96 (13.97) | 22 | 27 |
| K06 | 25°51″19.4″S 31°48″27.9″E | Komati River, SA | 252 | 21.76 | Wet | 1 (1.77) | 29.15 (47.97) | 81.83 (10.53) | 46 (2.65) | 138.33 (55.09) | 19.5 (5.26) | 6 | 4 |
| | | | | | Dry | 1 (1.73) | 50.31 (89.19) | 91.35 (8.32) | 64 (0) | 202.67 (45.23) | 58 (0) | 6 | 1 |
| L01 | 25°36″58.6″S 31°39″48.7″E | Lomati River, SA | 233 | 87.49 | Wet | 0.1 (0.32) | 3 (9.49) | – | 37 (0) | – | 30 (0) | 0 | 1 |
| | | | | | Dry | 0.71 (0.95) | 40.14 (68.72) | 76.52 (16.16) | 45.56 (0) | 131.5 (75.44) | 18 (0) | 4 | 1 |
| L02 | 25°37″53.1″S 31°39″19.0″E | Lomati River, SA | 236 | 89.69 | Wet | 0.1 (0.32) | 19.2 (60.72) | 95 (0) | – | 192 (0) | - | 1 | 0 |
| | | | | | Dry | 0 (0) | 0 (0) | – | – | – | – | 0 | 0 |
| L03 | 25°38″55.9″S 31°40″10.7″E | Lomati River, SA | 238 | 93 | Wet | 0 (0) | 0 (0) | – | - | – | – | 0 | 0 |
| | | | | | Dry | 0.2 (0.45) | 38 (84.97) | – | 98.24 (0) | – | 190 (0) | 0 | 1 |
| MB01 | 26°08″05.6″S 31°59″48.4″E | Mbuluzi River, SW | 163 | 23.14 | Wet | 4.5 (6.63) | 19.78 (24.01) | 62.65 (10.12) | 54.4 (12.35) | 53.65 (24.28) | 37.3 (29.32) | 17 | 10 |
| | | | | | Dry | 1 (1.41) | 11.19 (14.48) | 29.2 (7.05) | 27.33 (7.02) | 17.4 (12.19) | 15.33 (11.68) | 5 | 3 |
| MB02 | 26°10″00.5″S 31°53″50.7″E | Mbuluzi River, SW | 194 | 6.06 | Wet | 1 (1.41) | 24.21 (34.31) | 63.17 (18.76) | 58 (5.66) | 69.83 (42.83) | 41 (11.31) | 6 | 2 |
| | | | | | Dry | 1 (1.41) | 44.38 (56.99) | 49.2 (16.93) | 49 (1.41) | 120 (87.56) | 81 (5.66) | 5 | 2 |
| ML01 | 26°10″34.6″S 31°5″28.8″E | Mlawula River, SW | 147 | 47.5 | Wet | 1.57 (2.44) | 8.99 (12.56) | 49.29 (12.27) | 40.75 (3.59) | 30.43 (33.11) | 13 (3.92) | 7 | 4 |
| | | | | | Dry | 0 (0) | 0 (0) | – | – | – | – | 0 | 0 |
| ML02 | 26°11″16.4″S 31°59″12.4″E | Mlawula River, SW | 155 | 50 | Wet | 4 (3.67) | 21.97 (14.24) | 51.38 (8.39) | 49.91 (7.18) | 30.67 (16.48) | 27.09 (11.47) | 21 | 11 |
| | | | | | Dry | 0.86 (0.9) | 10.71 (11.22) | 29 (0) | 31.2 (6.30) | 13 (0) | 21.8 (14.06) | 1 | 5 |
| US01 | 26°46″57.5″S 31°59″04.3″E | Usutu River, SW | 79 | – | Wet | 0.8 (2.53) | 7.84 (24.78) | 71.8 (17.68) | 68.33 (9.87) | 85.2 (55.78) | 67 (25.24) | 5 | 3 |
| | | | | | Dry | 0.57 (1.13) | 11.43 (23.61) | 34.5 (9.75) | – | 28.50 (23.06) | – | 4 | 0 |
| US02 | 26°51″26.8″S 31°54″29.3″E | Usutu River, SW | 95 | – | Wet | 0.1 (0.32) | 2.2 (6.96) | 50 (0) | – | 17 (0) | – | 1 | 0 |
| | | | | | Dry | 0.14 (0.38) | 2.43 (6.43) | 31 (0) | – | 17 (0) | – | 1 | 0 |
| USCh | 26°50″51.0″S 31°54″49.8″E | Channel by Usutu, SW | 125 | – | Wet | 0.67 (1.16) | 22.67 (39.26) | 67 (14.14) | – | 68 (39.59) | – | 2 | 0 |
| | | | | | Dry | 0.29 (0.76) | 1.86 (4.91) | 26 (0) | 20 (0) | 11 (0) | 15 (0) | 1 | 1 |

Nunes et al. (2017), *PeerJ*, DOI 10.7717/peerj.3135

**Table 1** (*continued*)

| Site | Coordinates | Location | Elevation (m) | Distance to intro (km) | Season | CPUE (SD) (N/trap/night) | CPUE (SD) (g/trap/night) | Size M (SD) | Size F (SD) | Mass M (SD) | Mass F (SD) | M | F |
|------|-------------|----------|---------------|------------------------|--------|--------------------------|--------------------------|-------------|-------------|-------------|-------------|---|---|
| D01 | 25°33″08.1″S | Dam, SA | 190 | – | Wet | 15.3 (19.98) | 64.51 (35.17) | 66.34 (10.45) | 62.06 (10.78) | 78.39 (37.87) | 59.42 (34.24) | 118 | 35 |
| | 31°54″16.0″E | | | | Dry | 1.75 (2.12) | 46.87 (33.26) | 65.19 (12.02) | 59.5 (16.36) | 73.23 (48.46) | 50 (46.78) | 26 | 9 |
| D02 | 25°32″57.1″S | Dam, SA | 186 | – | Wet | 3.3 (2.83) | 40.89 (19.09) | 59.18 (9.62) | 53.86 (8.83) | 53.11 (26.12) | 36.67 (15.98) | 18 | 15 |
| | 31°53″37.0″E | | | | Dry | 1.9 (4.09) | 13.22 (23.20) | 56.12 (11.36) | 54.5 (8.42) | 45.8 (25.45) | 37.78 (16.11) | 10 | 9 |
| D03 | 25°37″14.4″S | Dam, SA | 190 | – | Wet | 2.67 (3.68) | 36.77 (49.89) | 66.71 (13.33) | 65.27 (14.28) | 84.95 (53.81) | 74.91 (49.93) | 21 | 11 |
| | 31°51″42.3″E | | | | | | | | | | | | |
| D04 | 25°32″41.2″S | Dam, SA | 188 | – | Wet | 1.11 (1.27) | 49.44 (52.00) | 69.57 (16.27) | 53.33 (15.04) | 102 (52.51) | 54 (6.93) | 7 | 3 |
| | 31°50″20.3″E | | | | Dry | 5.57 (6.45) | 42.48 (21.24) | 61.44 (8.49) | 51.4 (11.73) | 55.05 (27.89) | 33.47 (21.11) | 19 | 19 |
| D05 | 25°51″52.5″S | Dam, SA | 265 | – | Wet | 7.4 (8.93) | 16.05 (11.68) | 46.44 (9.97) | 42.15 (5.78) | 26.35 (15.97) | 17.36 (8.89) | 46 | 28 |
| | 31°50″00.9″E | | | | Dry | 0.9 (1.10) | 19.97 (26.92) | 50.97 (13.26) | 53.39 (10.99) | 36.8 (27.73) | 36 (15.41) | 5 | 4 |
| D06 | 25°58″43.6″S | Sand River Dam, SW | 295 | – | Wet | 0.13 (0.35) | 1.13 (3.18) | 38 (0) | – | 9 (0) | – | 1 | 0 |
| | 31°42″42.8″E | | | | Dry | 0.38 (0.74) | 5.38 (10.04) | 33 (0) | 27 (0) | 24.5 (0.71) | 13 (0) | 2 | 1 |

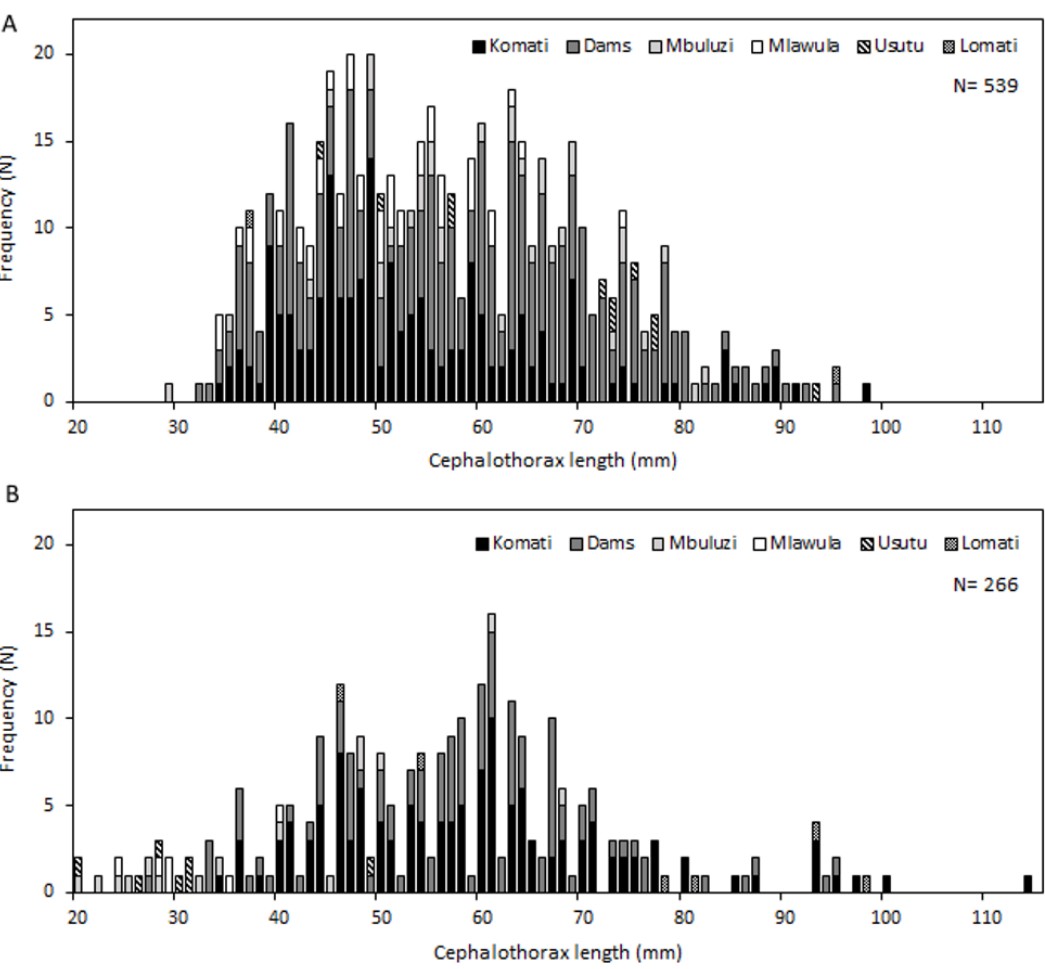

**Figure 3** Length-frequency distributions of *C. quadricarinatus* in different locations of the Komati, Mbuluzi, Mlawula, Usutu and Lomati rivers and in irrigation dams. (A) Wet season and (B) dry season.

Mlawula River (15.5 g trap night$^{-1}$ for wet season and 5.4 g trap night$^{-1}$ for dry season), Usutu River (10.9 g trap night$^{-1}$ for wet season and 5.2 g trap night$^{-1}$ for dry season) and in dams (34.8 g trap night$^{-1}$ for wet season and 25.6 g trap night$^{-1}$ for dry season) (Table 1).

## Size classes

Specimens of *C. quadricarinatus* varied widely in size, with cephalothorax lengths ranging from 20 to 114 mm, and individuals between 40 and 70 mm being by far the most numerous and representing 73% of all measured crayfish. Length-frequency graphs demonstrated the existence of multiple cohorts in the Komati, Mbuluzi, Mlawula and Usutu rivers, and also in irrigation dams. This did not seem to be the case for the Lomati River, where only very few size classes were present (Fig. 3).

Ovigerous females, or females carrying newly hatched crayfish (average size 63.8 mm, average mass 58.7 g) were found in October and December 2015, at five different sampling sites, three on the Komati River (K01, K02 and K03) and two in dams (D01 and D02)

(Table S2). The number of eggs ranged from 281 to 539 and the number of newly hatched crayfish ranged from 18 to 20 (many probably detached while in the traps).

### Sex ratio

In the wet season, the overall sex ratio (all sampling sites together) was significantly different from the expected sex ratio of 1: 1 ($\chi^2 = 58.856$, $p < 0.001$), with males outnumbering females, while this was marginally non-significant in the dry season ($\chi^2 = 3.626$, $p = 0.057$). Looking at specific areas, in the wet season, males were significantly more numerous than females in the Komati ($\chi^2 = 8.022$, $p = 0.005$) and Mlawula rivers ($\chi^2 = 3.930$, $p = 0.047$), as well as in dams ($\chi^2 = 45.478$, $p < 0.001$), but not in the Mbuluzi ($\chi^2 = 3.457$, $p = 0.063$) or the Usutu rivers ($\chi^2 = 1.600$, $p = 0.206$). In the dry season, sex ratios were not significantly different to the expected 1: 1 proportion ($p \geq 0.05$ in all cases). However, if we consider sampling sites individually, sex ratio was not significantly different from the 1: 1 proportion for most of them ($p > 0.05$ for most sites), except for sites K03 ($\chi^2 = 13.787$, $p < 0.001$), D01 ($\chi^2 = 45.026$, $p < 0.001$) and D05 ($\chi^2 = 4.378$, $p = 0.036$) in the wet season and D01 in the dry season ($\chi^2 = 8.257$, $p = 0.004$) (Table 1).

### Spread rate

In the Komati River, crayfish were found at a maximum distance of 112 km downstream of the point of introduction, indicating a mean downstream spread rate of 8 km year$^{-1}$ (using 2001 as the approximate year of first introduction). In the Lomati River, they were detected 93 km from the source of introduction, approximately 33 km upstream from the confluence with the Komati River. This indicates a total mean spread rate of 6.6 km year$^{-1}$ and, using the calculated mean spread rate of 8 km year$^{-1}$ downstream until the confluence with the Komati River, an upstream spread rate of 4.7 km year$^{-1}$.

### Variation with distance to source of introduction

No significant correlations were found between abundance, biomass or sex ratio of *C. quadricarinatus* during both wet and dry seasons, and distance to crayfish introduction source in the Komati River (for all correlations, $p > 0.05$). However, size and mass of both females and males was significantly correlated with distance to the source of crayfish introduction. Interestingly, a significant positive correlation was found between these variables for females in the wet season ($r = 0.344$, $N = 69$, $P = 0.004$ for size and $r = 0.438$, $N = 71$, $P < 0.001$ for mass), while during the dry season these correlations were negative ($r = -0.686$, $N = 63$, $P < 0.001$ for size and $r = -0.641$, $N = 63$, $P < 0.001$ for mass) (Fig. 4A). For males, the relationship was always negative, independent of season, but only statistically significant in the dry season ($r = -0.440$, $N = 66$, $P < 0.001$ for size and $r = -0.505$, $N = 66$, $P < 0.001$ for mass) (Fig. 4B).

## DISCUSSION

In this study we confirmed the presence of established and widespread populations of *C. quadricarinatus* in South Africa and Swaziland. Based on the evidence that populations have spread and are reproducing at multiple localities as far as 115 km from the point of

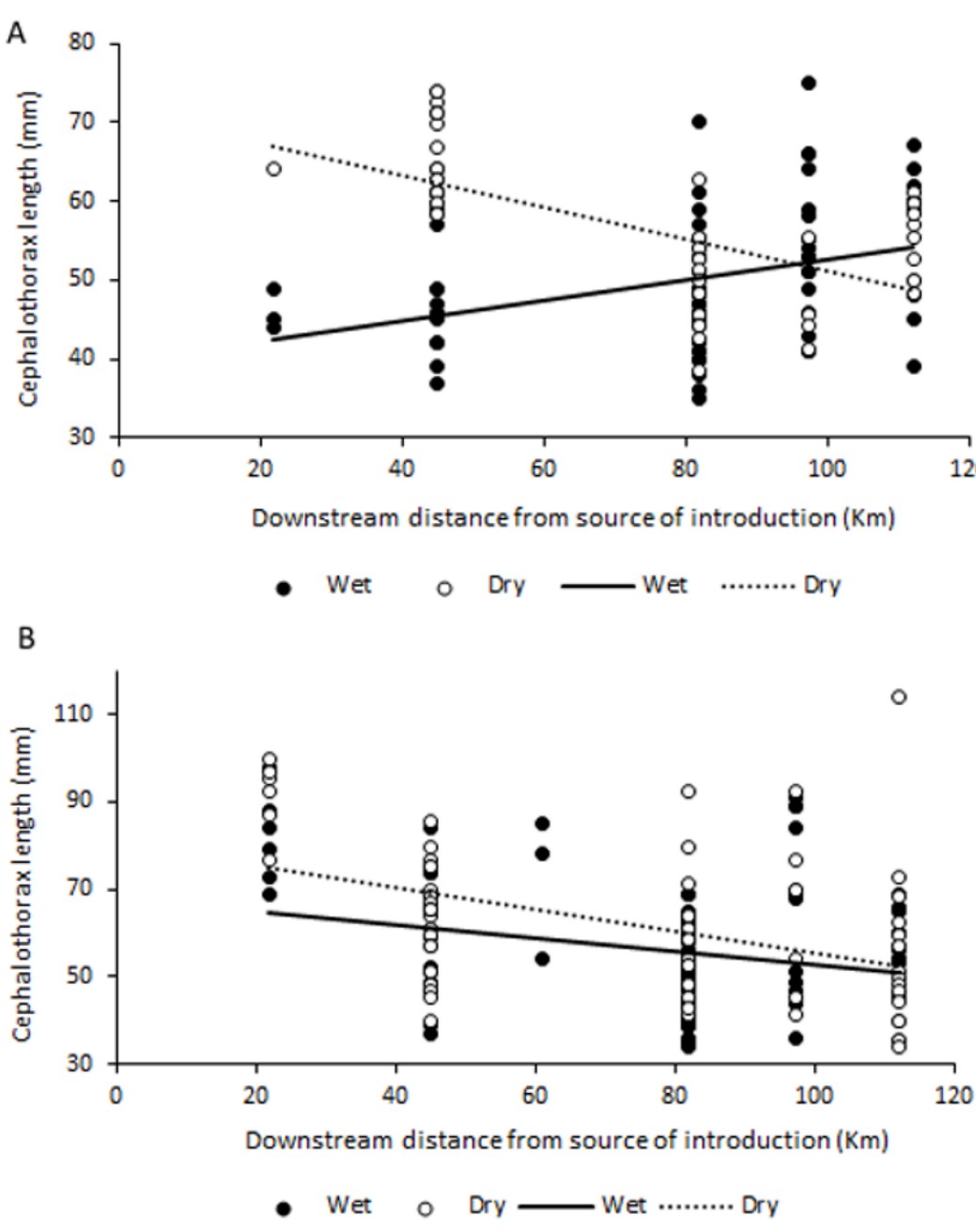

**Figure 4** Relationship between size (cephalothorax length, in mm) and distance to crayfish introduction source for *C. quadricarinatus* in the Komati River during the wet and dry seasons. (A) Females and (B) males.

introduction, this species can be considered as fully invasive (category E) in these countries, according to the criteria in *Blackburn et al. (2011)*. We also show how populations of this species have expanded in South Africa and Swaziland since they were first detected in 2002, being now present in at least three large rivers (Komati, Mbuluzi and Usutu), two tributaries (Lomati and Mlawula rivers), as well as in several irrigation dams. Crayfish populations were found to be established (presence of multiple cohorts and reproduction) at most

sampling sites, the main exception being the Lomati River, where very few individuals were sampled.

Although *C. quadricarinatus* were found to have dispersed upstream in two different tributaries (Lomati and Mlawula rivers), they were not detected upstream of the known point of introduction in the Komati River and in upstream sections of the Lomati River. This might be related with the large increase in elevation in these sampling points (274–433 m a.s.l.) and/or potential lower water temperatures. In the Lomati River, the Driekoppies Dam, located just by the border with Swaziland, and where no crayfish were found (or upstream of it), might also act as a dispersal barrier. Crayfish were also not detected in the Crocodile River; however, some specimens were recently detected approximately 10.7 km upstream of the furthest point sampled in this study (AC Hoffman & TA Zengeya, pers. obs., 2016). The fact that no individuals were sampled from the Ndumo Game Reserve was surprising and suggests small population sizes in the area, probably a result of an extended drought period.

Crayfish were not found in sites near Richard's Bay, indicating that the record from 2009 was indeed probably the result of an isolated introduction event, through release by aquarists or escape from an ornamental pond. This would not be surprising, as several crayfish species including *C. quadricarinatus* are available for sale in South Africa, either via online sources or in pet shops around the country (AL Nunes, pers. obs., 2016). The anecdotal reports of crayfish at Albert Falls Dam and Goedertrouw Dam could not be confirmed during the current surveys. However, it is important to note that, given the extensive size of these dams, it is extremely difficult to confirm crayfish absence, especially without an intensive and focused sampling, targeted specifically for these type of habitats.

Relative abundances of *C. quadricarinatus* in the Komati River (average 3.3 indv trap night$^{-1}$; maximum 9.4 indv trap night$^{-1}$) and in irrigation dams (average 3.7 indv trap night$^{-1}$; maximum 15.3 indv trap night$^{-1}$) were considerably higher than the ones found in other invasive populations of this species in Zimbabwe (maximum of 4.0 indv trap night$^{-1}$; *Marufu, Phiri & Nhiwatiwa, 2014*) and Slovenia (0.09 indv trap night$^{-1}$; *Jaklič & Vrezec, 2011*), reflecting how well the species has adapted in this region. In the Lomati River, crayfish were less abundant (average 0.2 indv trap night$^{-1}$), probably reflecting either a more recent invasion or a less suitable habitat (*Hudina et al., 2012*). The Lomati River is less regulated than the Komati River, containing fewer gauging weirs and consequently having higher flow velocity (AC Hoffman & AL Nunes, pers. obs., 2016).

The observed average size range of *C. quadricarinatus* collected in the various sampling sites (cephalothorax length: 20–98.2 mm) was in the range of values reported for this species other invasive populations (*Bortolini, Alvarez & Rodriguez-Almaraz, 2007*; *Jaklič & Vrezec, 2011*; *Marufu, Phiri & Nhiwatiwa, 2014*). The sex ratio at individual sampling sites was generally not significantly different from the commonly found 1: 1 ratio (e.g., *Bortolini, Alvarez & Rodriguez-Almaraz, 2007*; *Belle et al., 2011*). However, it is interesting that the overall sex ratio (all sampling sites) in the wet season was significantly different from 1:1, with males outnumbering females. This probably reflects reduced capture vulnerability of females during the reproduction season, when berried females are less active (*Masser & Rouse, 1997*).

The species exhibited potential to disperse both downstream of the different initial invasion points and upstream of two different tributaries. In the Inkomati basin, downstream and upstream spread occurred at a rate of 8 and 4.7 km year$^{-1}$, respectively. However, the downstream rate might be higher, considering the high likelihood that the species has already spread further downstream in the Komati River into the Mozambican side (which could not be sampled in this study). *Cherax quadricarinatus* most likely reached the Mbuluzi River basin via irrigation canals that act as an interbasin water transfer between the Mbuluzi and Inkomati basins (similarly to that facilitating the spread of an alien loricariid catfish in the KwaZulu-Natal province; *Jones et al., 2013*). While the date of introduction is uncertain, crayfish were observed for the first time in 2009 at the Pequenos Libombos Dam in southern Mozambique (I Nerantzoulis, pers. comm., 2016), and were recorded as established in 2011 (Fig. 2B; *Chivambo, Nerantzoulis & Mussagy, 2013*). Assuming this was the result of natural spread, and not of an exceptional translocation event, this demonstrates that in eight years, and in a downstream direction, the species covered 40 km of channels between the Mbuluzi and Inkomati basins, plus 76.8 km in the Mbuluzi River until the Pequenos Libombos Dam, indicating a potential mean spread rate of 14.6 km year$^{-1}$.

Down and upstream dispersal have been observed for other invasive crayfish species, ranging from 1.8 to 24.4 km year$^{-1}$ (downstream) and 0.35 to 4 km year$^{-1}$ (upstream) for *Pacifastacus leniusculus* in different European countries (*Bubb, Thom & Lucas, 2005*; *Hudina et al., 2009*; *Weinländer & Füreder, 2009*; *Bernardo et al., 2011*), 0.5 to 3.10 km year$^{-1}$ (upstream) for *P. clarkii* (*Bernardo et al., 2011*; *Ellis et al., 2012*), and 12 to 84 km year$^{-1}$ (downstream) and 2.5 km year$^{-1}$ (upstream) for *Orconectes limosus* in Eastern Europe (*Hudina et al., 2009*). This indicates that the first estimates of dispersal rates for *C. quadricarinatus*, especially for upstream movements, are high, once again suggesting a high invasion potential of the species in the study area. Furthermore, irrigation dams, where crayfish populations seem to become very abundant, might act as secondary sources of crayfish invasions or as stepping stones for range expansion through irrigation channels or over land, facilitating subsequent establishment in new irrigations dams, rivers or tributaries.

In the Komati River, which has been colonised for the longest time, crayfish were generally larger and heavier close to the initial introduction point, with sizes decreasing as distance to the invasion source increased. A similar pattern has been observed for round goby invasions in Canada (*Ray & Corkum, 2001*; *Brownscombe & Fox, 2012*) and the same tendency found for signal crayfish in Croatia (*Hudina et al., 2012*), suggesting that juveniles may disperse more actively and rapidly than adults, likely due to high intraspecific competition. In the case of females, this might also indicate a strategy that allocates resources to favour reproduction with increased offspring closer to the source, as egg number is a function of female size (*Jones, 1990*). It is important to note that as distance to the invasion source increases in the Komati River, elevation decreases. This means that the pattern found may also indicate that larger and heavier individuals are more common in more elevated areas.

However, the opposite pattern was observed for females during the wet season, with smaller females found near the introduction point and larger ones further downstream.

Given that sexual maturity is generally reached when animals attain around 50–60 g (*Jones, 1990*), corresponding to approximately 55–65 mm cephalothorax length in this study, this may indicate that mature females might be reproducing at different times of the year along the invasion gradient. In sites further away from the source females are spawning in October–December (and perhaps repeatedly), while reproduction might be taking place at a different time of the year in longer established populations. Nevertheless, taking into account that large berried females are usually less active and, therefore, less susceptible to capture (*Masser & Rouse, 1997*), differences in reproductive activity might be affecting sampling efficiency. Still, the possibility that reproductive behaviour might differ along the invasion gradient warrants further investigation, especially considering that *C. quadricarinatus* has a natural reproductive season throughout spring and summer, with spawning occurring more than once from October to March (*Jones, 1990*; *Masser & Rouse, 1997*). Alternatively, the pattern found might also suggest that large females closer to the invasion front are more active and disperse during the wet season, which might contribute to further range expansion (*Brownscombe & Fox, 2012*).

Although current legislation prohibits the importation, release and movement of *C. quadricarinatus* in South Africa (*Republic of South Africa, 2016*), the lack of resources (both manpower and financial) makes it extremely challenging to enforce these regulations. Furthermore, taking into account the accidental escape of *C. quadricarinatus* from an aquaculture farm in Swaziland and consequent spread to South Africa and Mozambique, this study reinforces the importance of putting international agreements regarding invasive species into practice. The SADC Protocol on Fisheries, for example, prohibits the introduction of alien species into aquatic ecosystems shared by two states, unless all the affected states agree to the introduction (*De Moor, 2004*). Clearly, there is a need to strengthen and better coordinate the enforcement and effectiveness of existing protocols between neighbouring countries in Africa, in what concerns introduction and spread of invasive species. Taking into account that, once established, invasive crayfish populations are usually impossible to eradicate, transnational cooperation should also be taken into account regarding possible management actions (e.g., mechanical, physical, chemical and/or autocidal methods; reviewed in *Gherardi et al., 2011*) to contain or hinder the spread of *C. quadricarinatus* in these international river systems. These actions would need to be implemented by all countries involved (South Africa, Swaziland and Mozambique), in order for the efforts of one country to not be jeopardised by the other non-complying countries.

## CONCLUSIONS

This study shows that populations of *C. quadricarinatus* are now established and spreading in South Africa and Swaziland. While the environmental impact of *C. quadricarinatus* in newly invaded habitats has yet to be determined, local communities in South Africa have already started harvesting it (*Coetzee et al., 2015*), increasing the risk of translocations for commercial reasons. The possible introduction of this species into new catchments in Africa is a matter of extreme concern, especially given the high speed at which the species has been expanding and its potential impacts on native biota, such as disease introductions,

competitive interactions with native freshwater crustaceans or habitat modifications (*De Moor, 2002*; *Nunes et al., 2016*). However, as no formal research has been done on the impacts of *C. quadricarinatus* invasive populations in any part of the world, the species would be classified as 'Data Deficient' (current information insufficient to assess level of impact) according to *Blackburn's et al.* (*2014*) environmental impact classification for alien taxa. This calls for an immediate assessment of potential impacts of this species on native freshwater ecosystems in Africa.

## ACKNOWLEDGEMENTS

We are greatly indebted to Len Coetzer for showing us various sampling sites in South Africa and for providing us the contact details of numerous farmers and researchers in the area. We also thank Vhutali Nelwamondo, Jonathan Vervaeke and Rheul Lombard for their invaluable help during field work. We are grateful to SANParks, in the person of Robin Petersen, for permission to sample the Crocodile River inside Kruger National Park. We are grateful to Prof. CN Magagula for her essential help in contacting permitting authorities in Swaziland and to Morgan Vance at the Savannah Research Centre for receiving us so well in Swaziland. ALN, OLFW, TAZ and GJM thank the National Research Foundation and the DST-NRF Centre of Excellence for Invasion Biology for their continued support.

### Funding

This study was funded by the South African National Department of Environment Affairs through the South African National Biodiversity Institute Invasive Species Programme. OLFW (Grant No. 77444) and TAZ (Grant No. 103602) have received funding from the National Research Foundation. The funders had no role in study design, data collection and analysis, decision to publish, or preparation of the manuscript.

### Grant Disclosures

The following grant information was disclosed by the authors:
South African National Department of Environment Affairs.
National Research Foundation: 77444, 103602.

### Competing Interests

John Measey is an Academic Editor for PeerJ.

### Author Contributions

- Ana L. Nunes conceived and designed the experiments, performed the experiments, analyzed the data, contributed reagents/materials/analysis tools, wrote the paper, prepared figures and/or tables, reviewed drafts of the paper.
- Tsungai A. Zengeya conceived and designed the experiments, performed the experiments, contributed reagents/materials/analysis tools, reviewed drafts of the paper.
- Andries C. Hoffman performed the experiments, contributed reagents/materials/analysis tools, reviewed drafts of the paper.

- G. John Measey and Olaf L.F. Weyl conceived and designed the experiments, contributed reagents/materials/analysis tools, reviewed drafts of the paper.

## Field Study Permissions

The following information was supplied relating to field study approvals (i.e., approving body and any reference numbers):

Permits were obtained from the Mpumalanga Tourism and Parks Agency (MPB. 5523), Ezemvelo KZN Wildlife (OP 4428/2015) and Mbuluzi Game Reserve.

## Data Availability

The raw data has been supplied as Datas S1 and S2.

## Supplemental Information

Supplemental information for this article can be found online at http://dx.doi.org/10.7717/peerj.3135#supplemental-information.

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
