# Peer review of "Distribution and establishment of the alien Australian redclaw crayfish, Cherax quadricarinatus, in South Africa and Swaziland"

_PeerJ, doi:10.7717/peerj.3135_

## Round 0.1 · original submission · Minor Revisions

Dear Authors,

I have just received the comments of the 2 referees who are both positive about the manuscript and agree that only minor revisions are needed in order to make your manuscript publishable in PeerJ.

I look forward to seeing the revised version of your manuscript

·

Basic reporting

The article is well written, English is professional.

The article has sufficient introduction and relevant references. Minor comments are labeled in the attached pdf.

The structure, figs and tables are professional and relevant.

Raw data are shared as supplement.

The article represents an appropriate "unit of publication".

Experimental design

The research is original and consistent with journal's aims and scope.

The question is well-defined, because invasion of non-indigenous crayfish species is a problem highlighted in many countries. Monitoring of invasive species and prediction of their spread are very important for wildlife management and conservation of native species and entire ecosystems as well. The findings in the article are helpful in this regard.

There are no conflicts with ethical standards and investigation and technical standard are suitable.

Methoddological section is described with sufficient information to be reproducible.

Validity of the findings

Data are robust, samplings covered complete season and many localities. Statistical analysis is adequate.

Conclusion is well stated and connected with original research.

Additional comments

Dear authors,

I reviewed the MS entitled "Distribution and establishment of the alien Australian redclaw crayfish, Cherax quadricarinatus, in South Africa and Swaziland". The paper is well writen, clear and focused on invasive behaviour of redclaw. This is a helpful addition to our knowledge about invasion of this crayfish species and is usable for wildlife management, policymakers and other stakeholders. Therefore, I recommend the paper for publication in PeerJ after minor revision.

I suggest to add analysis of spread based not only on the distance from point of introduction but also on elevation (slope). Other suggested minor changes are labeled in the attached pdf.

Sincerely,
Jiri Patoka

Reviewer 2 ·

Basic reporting

The manuscript entitled: "Distribution and establishment f the alien Australian redclawed, Cherax cuadricarinatus in South Africa and Swaziland by Nunes et al. deals a monitoring study focus in detecting the presence of this Native Australian freshwater species in two South African countries. The manuscript is well written and the ideas clearly exposed. The literature reference is suitable and the background of the study correctly presented.
The study clearly shows the presence, establishment and spread of this species in the area of study. This finding in itself is an important finding alerting of the presence and spread of a species with great potential for causing negative impacts in the African ecosystems. The data are well analyzed and the conclusions to correct. I therefore recommend this work to be accepted for publication provided that some minor changes are provided.

Experimental design

The experimental design is simple but correct and sufficient. The research question is well defined except in the lines 96-97. In these lines, the authors mention that they have further investigated the characteristic, abundance, biomass, sex ratio, body size and mass with distance of the introduction. However, these aspects are poorly defined in the material methods which need to be better structured in blocks regarding each issue investigated rather than in only 2: Field study and Study area. This also needs to be better structured in sub-sections in the results section according to material methods.

Validity of the findings

The results are novel, data is robust, and conclusions clear. The findings are, therefore, valuable since they show the presence, establishment and spread, of an invasive species with negative potential impacts in the African ecosystems.
However, some of the discussion needs to be done on the basis of the data found and some conclusions regarding abundance, biomass, sex ratio, body size and mass need to be better explained in Material and Methods and results in order to better discuss the findings. The discussion is a bit long and I would suggest making it at least one page shorter.
Conclusions are fine regarding raising concern of the potential negative effects and the need to provide data for classification of this species as invasive. However, statements regarding the main findings of this work are missing or not stressed.

Additional comments

Well written manuscript reporting an invasive species with great potential damage in African, Minor changes are suggested.

---

## Round 0.2 · accepted · Accept

Thank you for submitting a revised version of your manuscript. I am pleased to inform you that your ms has been accepted for publication in PeerJ